# Microbial Detection and Quantification of Low-Biomass Water Samples Using an International Space Station Smart Sample Concentrator

**DOI:** 10.3390/microorganisms11092310

**Published:** 2023-09-13

**Authors:** Adriana Blachowicz, Camilla Urbaniak, Alec Adolphson, Gwyn Isenhouer, Andy Page, Kasthuri Venkateswaran

**Affiliations:** 1Biotechnology and Planetary Protection Group, Jet Propulsion Laboratory, California Institute of Technology, Pasadena, CA 91109, USA; ada.blachowicz@jpl.nasa.gov (A.B.); camilla.urbaniak@jpl.nasa.gov (C.U.); 2ZIN Technologies Inc., Middleburg Heights, OH 44130, USA; 3InnovaPrep LLC, Drexel, MO 64742, USA; aadolphson@innovaprep.com (A.A.); gisenhouer@innovaprep.com (G.I.); apage@innovaprep.com (A.P.)

**Keywords:** International Space Station, water concentrator, microbial monitoring, microgravity, molecular biology

## Abstract

The pressing need to safeguard the health of astronauts aboard the International Space Station (ISS) necessitates constant and rigorous microbial monitoring. Recognizing the shortcomings of traditional culture-based methods, NASA is deliberating the incorporation of molecular-based techniques. The challenge, however, lies in developing and validating effective methods for concentrating samples to facilitate this transition. This study is dedicated to investigating the potential of an ISS Smart Sample Concentrator (iSSC) as an innovative concentration method. First, the iSSC system and its components were tested and optimized for microgravity, including various testing environments: a drop tower, parabolic flight, and the ISS itself. Upon confirming the system’s compatibility with microgravity, we further evaluated its proficiency and reliability in concentrating large volumes (i.e., 1 L) of water samples inoculated with different microbes. The samples carried 10^2^ to 10^5^ colony-forming units (CFUs) of *Sphingomonas paucimobilis*, *Ralstonia pickettii*, or *Cupriavidus basilensis* per liter, aligning with NASA’s acceptable limit of 5 × 10^4^ CFU/L. The average retrieved volume post-concentration was ≈450 µL, yielding samples that were ≈2200 times more concentrated for subsequent quantitative PCR (qPCR) and CFU analysis. The average microbial percent recovery, as assessed with CFU counts, demonstrated consistency for *C. basilensis* and *R. pickettii* at around 50% and 45%, respectively. For *S. paucimobilis*, the efficiency oscillated between 40% and 80%. Interestingly, when we examined microbial recovery using qPCR, the results showed more variability across all tested species. The significance of these findings lies not merely in the successful validation of the iSSC but also in the system’s proven consistency, as evidenced by its alignment with previous validation-phase results. In conclusion, conducted research underscored the potential of the iSSC in monitoring microbial contamination in potable water aboard the ISS, heralding a paradigm shift from culture-based to molecular-based monitoring methods.

## 1. Introduction

Preserving the health and safety of the astronauts aboard the International Space Station (ISS) remains a top priority; the space station, with a rotating crew of three to ten individuals, necessitates continual and vigilant microbial monitoring [1,2]. Particularly, maintaining the quality of potable water onboard is of paramount importance, as it significantly impacts daily crew activities such as drinking, food rehydration, hygiene, and addressing medical emergencies [3,4].

Given this importance, the ISS Medical Operations Requirements Documents mandate that potable water onboard should not exceed a microbial concentration of 5 × 10^4^ CFUs per liter [5]. The prevalent microbial monitoring protocols rely on a culture-based approach, which carries the risk of not detecting non-cultivable microbes possibly present in the water [6]. As we begin considering a transition to molecular methods, the challenge lies in developing a rapid, effective concentration system to handle large-volume samples (>1 L).

At present, the so-called “gold standard” for concentration methods onboard the ISS is filtration through a membrane filter [7]. The process, though effective, has its limitations, such as a lack of automation, extended concentration times for large volume samples (>1 L), and the dependency of the results on the diameter of the used filter (the larger filter sizes shorten the processing time, but compromise the final concentration factor, when compared to smaller sizes). These issues underscore the urgent need for a microgravity-compatible filtration system capable of concentrating large-volume samples.

Addressing this need, the NASA Small Business Innovation Research (SBIR) program funded a three-year project proposed by InnovaPrep LLC. While the company’s Concentrating Pipette CP-150 and CP Select have demonstrated success in concentrating environmental samples retrieved from the ISS [8] and other low biomass environments like spacecraft assembly facilities [9], significant innovation was required for effective use in microgravity environments. Under the auspices of the SBIR project, InnovaPrep developed the lightweight, compact, and self-contained ISS Smart Sample Concentrator (iSSC) (Appendix A), capable of processing large volumes of the sample (up to 1 L) in less than five minutes. Additionally, this system could be easily scaled up for larger samples and could likely run 5 L in less than 15 min.

Phase II of the SBIR project saw the comparison of the iSSC system with the commercially available CP-150 and the Millipore filtration system, using bacterial species isolated from the ISS’ potable water: *Sphingomonas paucimobilis*, *Ralstonia pickettii*, and *Cupriavidus basilensis* [2,10,11,12]. One-liter samples containing 10^4^ CFU/L of each species were concentrated using the three concentration methods. The iSSC significantly outperformed the traditional filtration/concentration methods, boasting a concentration factor up to 15,000 times and improved bacteria capture and recovery efficiency [13].

The present SBIR Phase III study delves into assessing the efficacy and reproducibility of the iSSC system with various bacterial concentrations, spanning from 10^2^ to 10^5^ CFU/L. The study employed both traditional culture- and molecular-based techniques to determine efficiency. Furthermore, it assured reproducibility by utilizing biological (*n* = 3) and technical replicates (*n* = 9). The results obtained from this phase, focusing on the commercialization of the innovative technologies, products, and services resulting from the Phase II contract, were compared with those from Phase III, which centered on the development, demonstration, and delivery of the proposed innovation.

## 2. Materials and Methods

### 2.1. The ISS Smart Sample Concentrator (iSSC) Technology 

Grounded on the success of previously deployed and commercialized technology—the CP-150 and its enhanced version, the CP Select model, InnovaPrep [14]. ventured into creating the iSSC system (Appendix A), which is suitable for use in microgravity. Central to the CP-150 and CP Select are Concentrating Pipette Tips (CPTs), hollow-fiber membrane filters that enable capturing of microbes from samples containing up to 5 L. Once these microbes are captured, they are carefully eluted using InnovaPrep’s distinctive wet foam elution process.

The elution process leverages a buffered elution fluid that contains Tween 20, which acts as a foaming agent. Stored under a nominal head pressure of 125 psi of carbon dioxide, the fluid is held in small aerosol canisters. When the time for elution arrives, this fluid is released through a timed valve into the atmospheric pressure, resulting in carbon dioxide forming microbubbles in the solution. Due to the volume of the carbon dioxide that the elution fluid contains, the resultant foam expands to roughly 6 times the original fluid volume. This expanded wet foam then carries captured microbes, yielding a highly concentrated liquid sample ranging from 0.2 to 0.5 mL in volume.

However, for the iSSC system to be compatible with the microgravity environment, it had to undergo several crucial alterations. Key adaptations included (i) incorporating a sample bag, (ii) adding a bladder bag nestled in a vacuum bottle to hold the processed sample, (iii) employing a bag-on-valve (BOV) aerosol canister to manage the elution fluid before release, and (iv) creating a detachable capillary flow assisted container (CFAC) for better control of the concentrated sample (Figure 1A–D adapted from [13]. Within the core of the system, the iSSC assembly contains a concentration cell (CC) which harnesses hollow-fiber membrane filters to retain microbes during processing of the entire sample (Figure 1C).

A diaphragm pump was included in the design to create negative pressure on a vacuum bottle containing a bladder bag, allowing the sample to be drawn into the system. Upon completion, this pump is reversed to dispose of the waste from the processed sample, maintaining control over the sample even in microgravity.

In this unique process, the entire 1-L water sample is directed through the hollow-fiber concentration cell. The sample fluid navigates the lumens of seventy-two 0.2 µm polysulfone hollow-fiber membrane filters and, in post-processing, finds its way into the bladder bag. Particles larger than the pore sizes are captured in the fiber lumens and are eluted into a concentrated volume via the wet foam elution process.

While the use of elution fluid in a standard aerosol canister is common on Earth, for usage in microgravity, a method to control the fluid and prevent the release of gaseous carbon dioxide had to be innovated. Thus, the iSSC adopts a small volume of carbonated elution fluid, housed in a silicone “bag”, which is attached to the aerosol valve and contained within the Hand-Held Elutor (HHE) aerosol canister (Figure 1C). Akin to commercially available BOV aerosol devices, these HHEs are manufactured by injecting gas between the BOV and the aerosol canister wall before filling the BOV with a set volume of elution fluid. This method allows a small set volume of carbonated elution fluid to be stored and later released in a single burst.

Before the elution process, the iSSC diaphragm pump is used to create negative pressure in the CFAC. With the actuation of the HHE valve, the wet foam travels through the hollow-fiber membrane-filter lumens and into the CFAC. Here, the low-pressure environment causes the foam to break down rapidly and completely. As the elution fluid is released from the pressurized carbon dioxide environment of the HHE to a low-pressure environment, carbon dioxide escapes from the fluid in the form of microbubbles and a viscous wet foam is formed. During and after traveling through the fiber lumens, film thinning, bubble coalescence, and bubble bursting create instability and eventually the complete failure and breakdown of the foam. These mechanisms for carbonated wet foam are not influenced by gravity, thus the processes occurring in microgravity and 1G will occur at a similar rate.

Once the foam has broken down, the CFAC design uses capillary forces to passively separate the fluid phases. Through the utilization of surface tension, wetting conditions, and container geometry, the liquid is delivered to a port for withdrawal. Using a repeater pipette or another transfer device, the sample can be transferred to a molecular assay for the detection of target organisms.

### 2.2. Bacterial Cultures and Growth Conditions

An outline of the experimental procedure for Phase III of the iSSC validation study can be found in Figure 2. *S. paucimobilis*, *R. pickettii*, and *C. basilensis*, bacterial species isolated from the ISS potable water, were kept in glycerol stocks at −80 °C for preservation [2,10,11]. For each experiment, isolates were streaked onto tryptic soy agar TSA) from the glycerol stocks and were incubated at 30 °C for 48 h. After incubation, a bacterial suspension was prepared in a sterile phosphate saline buffer (PBS, pH 7.4). The suspension density was measured using Densicheck (bioMérieux, St. Louis, MO, USA) and then diluted to densities of 10^5^, 10^4^, 10^3^, and 10^2^ CFU/mL. The bacterial suspensions, also referred to as the “inoculum”, were added to 1-L volumes of sterile PBS in triplicate. This resulted in unconcentrated samples with densities of 10^5^, 10^4^, 10^3^, and 10^2^ CFU/L.

### 2.3. Sample Concentration

Immediately after preparation, the unconcentrated samples of *S. paucimobilis*, *R. pickettii*, and *C. basilensis* were processed and concentrated with the 0.2 μm hollow fiber polysulfone iSSC filtration system. The captured microbes were then released using the novel wet foam elution system. Subsequently, these concentrated samples were utilized for culture-based and molecular-based analyses.

### 2.4. Culture-Based Assessment after Concentration

Post-concentration, the samples underwent appropriate serial dilutions in sterile PBS. From each of these diluted samples, 100 µL was plated onto tryptic soy agar plates (TSA) in duplicate, and then incubated for 48 h at 30 °C. The original inoculum was plated similarly. The recovery percentage from the iSSC concentrator was calculated by dividing the CFU counts of the concentrated samples by the CFU counts of the original inoculum. The data was visualized using GraphPad Prism version 9 (GraphPad Software, La Jolla, CA, USA). 

### 2.5. qPCR Assessment after Concentration

DNA was extracted from the inoculum and the concentrated samples using the Maxwell 16 Tissue LEV Total RNA purification kit, following the manufacturer’s instructions for the Maxwell 16 automated system (Promega, Madison, WI, USA). The DNA was then eluted in 50 µL of molecular-grade water and preserved at −20 °C. For qPCR analysis, the extracted DNA was used, targeting the species-specific single-copy gene *gyrB* with the QuantStudio 6 Flex Real-Time PCR System (ThermoFisher Scientific, Waltham, MA, USA). This gene encodes for the B subunit of the DNA gyrase and serves as a phylogenetic marker in bacterial species identification [12,14]. Table 1 summarizes the designs of the primers and probes targeting the gyrB gene, along with the standards for each tested species. Each qPCR reaction included species-specific probes, forward and reverse primers, and template DNA. Samples were run in triplicate. Each reaction was carried out in the following conditions: initial denaturation at 95 °C for 4 min, followed by 40 cycles of denaturation at 95 °C for 30 s and a combined annealing and extension at 55 °C for 35 s. The number of gene copies in the samples was determined by running a standard curve, which was generated using serial dilutions (10^1^–10^5^) of the synthesized species-specific *gyrB* gene. The no-template controls were not subtracted. The concentration efficiency of each concentrated sample was calculated by dividing the copy number of the concentrated sample by the copy number of the unconcentrated inoculum.

### 2.6. Statistical Analyses

Figures and statistical analyses were generated using GraphPad Prism version 9 (GraphPad Software, La Jolla, CA, USA). Statistical analysis included 1-way ANOVA, t-test with Welch’s correction and significance was based on *p* < 0.05. 

## 3. Results and Discussion

### 3.1. The ISS Smart Sample Concentrator (iSSC) Optimization for Microgravity

The iSSC underwent rigorous testing to ensure its safe and efficient operation in microgravity environments, specifically targeting its capabilities in concentrating bacteria reliably and effectively. The microgravity performance of the iSSC system and its components were assessed through three primary strategies: (i) drop tower testing of the CFAC; (ii) parabolic flight testing of the complete iSSC setup; and (iii) an ISS flight test of the CFAC monitored by the WetLab-2 team. Each test was specially tailored to evaluate the operation of the iSSC, except for the WetLab-2 team test, which was conducted to contemplate the replacement of the current WetLab-2 debubbler with the CFAC.

(i) The CFAC underwent drop tower testing at the Dryden Drop Tower, located at Portland State University, conducted by IRPI LLC (Appendix A). As shown in Figure 3B, the sample was observed wicking to the CFAC outlet port (located at the top) at 0 G, as well as a 1 G control. These tests reflected exceptional performance, predicted to deliver bubble-free samples at the CFAC outlet in microgravity environments. It’s important to underscore that results obtained in 0 G are relevant and applicable to microgravity environments as well [15,16,17,18].

(ii) The complete iSSC system was put to the test for its microgravity performance through a 30-parabola flight test conducted by the Zero Gravity Corporation on 21 March 2017. The testing procedure was divided across consecutive parabolas. Startup and concentration were addressed during the first parabola, while sample depletion and sample bag operation took place in the second. Sample elution, including breakdown of the wet foam, occurred during the third parabola, and withdrawal of the eluted sample was carried out during the fourth parabola. These procedures successfully validated all iSSC operations in a microgravity environment (Figure 3C).

(iii) A two-series test run of the CFAC was conducted aboard the ISS (Figure 3D). In the first series, dye-colored buffer solutions were extracted from partially filled CFAC devices and dispensed into Smart Cycler tubes to validate control over these solutions at the CFAC sample port. In the second series, DNA solutions were extracted from partially filled CFACs and then dispensed into Smart Cycler tubes containing lyophilized reagents for analysis. The quality of the data from three replicate tubes was verified using a qPCR curve (Appendix A). These results provided substantial evidence that the CFAC could replace the current WetLab-2 pipette loader, thereby reducing crew time requirements and lowering up-mass and overall costs. It is noteworthy to mention that these experiments were performed using buffers that lacked surfactants, while the iSSC elution fluid contained 0.075% Tween 20. Therefore, these results provide indirect evidence supporting the predicted performance of the CP Select™ as part of the iSSC in a microgravity environment.

### 3.2. Recovery Volume

The Phase III iSSC prototype validation revealed average volumes of 449 μL, 491 μL, and 428 μL, recovered for *S. paucimobilis*, *C. basilensis*, and *R. pickettii*, respectively (Table 2). After analysis with culture and qPCR, the elution volume of each sample was considered for the computation of the total microbial load. It is worth noting that the average volume retrieved in the Phase II iSSC system validation was 304 μL, whereas the Millipore system provided an average of 928 μL. The Phase III study underscored a twofold increase in the elution-based concentration factor of the iSSC system in comparison to the Millipore system assessed during the Phase II validation (Table 3).

### 3.3. Percent Recovery Based on CFU Counts

Figure 4 and Appendix A depict the efficiency of the iSSC system in recovering three bacterial strains isolated from the ISS. The acceptable microbial limit for drinkable water is 50 CFU/mL, thereby prompting an efficiency evaluation of the iSSC system at 10^4^ CFU/L and two logs lower. At 10^4^ CFU/L, recoveries of 70%, 50%, and 50% were recorded for *S. paucimobilis*, *C. basilensis*, and *R. pickettii*, respectively. Both *C. basilensis* and *R. pickettii* demonstrated consistent average recovery rates across all tested concentrations, at about 50% and 45%, respectively. In contrast, *S. paucimobilis* exhibited a rising efficiency, ranging from 40% to 80%. The recovery variability in *S. paucimobilis* post-filtration among the different concentrations may have been influenced by the use of high-vacuum grease to seal the iSSC HHE consumable. Importantly, despite the variable recovery rates across the tested microorganisms, no statistically significant changes were observed.

### 3.4. Percent Recovery Based on qPCR Data

Figure 5 and Appendix A detail the assessment of the iSSC system’s microbial recovery efficiency using a molecular method for *S. paucimobilis*, *R. pickettii*, and *C. basilensis*. The *gyrB* gene copy numbers [19] were quantified for each strain at various microbial cell concentrations (10^2^ to 10^5^ cells/L). The copy number efficiency was expressed as a percentage, calculated by dividing the *gyrB* gene copy number of the concentrated sample by that of the inoculum. Notably, the qPCR method displayed more variability in bacterial recovery efficiency than culture-based approaches. However, the overall detected *gyrB* copies matched the expected microbial cell concentrations. Interestingly, around 18% of the samples showed values exceeding 100% when evaluated via qPCR but not the CFU method. This anomaly was only observed in low biomass samples, warranting further investigation. One possible reason might be related to the variable DNA extraction efficiency for low biomass samples. It is also worth mentioning that in this study dead and alive cells were not distinguishable via qPCR as no intercalating dye like propidium monoazide (PMA) was added to the concentrated samples.

As NASA contemplates transitioning to molecular-based methods for microbial monitoring aboard the ISS, understanding potential issues is vital. The study shows that the iSSC system effectively concentrates one-liter water samples and detects bacteria at concentrations as low as 10^2^ CFU/L using qPCR. Future research will focus on assessing only viable microorganisms as we transition to DNA-based microbial monitoring systems.

### 3.5. Phase II vs. Phase III Culture-Based Results Comparison

The Phase II iSSC system validation focused on recovery rates at a single concentration of 10^4^ CFU/L [13]. Hence, the Phase III results were compared with the Phase II results at a starting inoculum of 10^4^ CFU/L to check for consistency. Both validation phases showed comparable recovery rates for *S. paucimobilis* and *R. pickettii* (~60% and ~50%, respectively) at the 10^4^ CFU/L concentration, while *C. basilensis* exhibited a 20% drop in the Phase III recovery rate (Figure 6). These variations were not statistically significant, suggesting they might be due to differing physiological responses to the distinct iSSC units used in the two phases. Alternatively, in Phase III, the CFAC design was altered to make it more user-friendly. The CFAC was altered to be removable, therefore making it easier to collect and store the sample if needed. This design change may or may not have affected the recovery efficiency; however, it was beyond the scope of this project to rebuild and retest the filter cells in the previous configuration. Irrespective of the tested iSSC unit during Phase III, these results confirmed the superior performance of the iSSC system when compared to the Millipore filtration system, which showed only 10–20% microbial recovery (*p* = 0.0004 and *p* = 0.0155 for *S. paucimobilis* and *R. picketii*, respectively). With that recovery rate in mind, it is plausible that syringe-based filtration methods may not be suitable to measure lower microbial abundance (10^2^ to 10^3^ CFU/L), which was successfully detected using the microgravity-compatible iSSC system.

The iSSC system’s compact size, lightweight design, and high concentration factors (~2200×) make it compatible for space travel, unlike the Millipore system, which requires a vacuum pump and additional accessories. This makes the iSSC system a viable alternative to the current syringe-based filtration techniques used aboard the ISS. The iSSC system can filter larger volumes (>1 L), which is not possible with the current method, which is limited to 10 mL, while at the same time, the concentrated samples obtained through the iSSC system can be used for versatile downstream analyses, such as DNA extraction, qPCR analysis, and sequencing, with MinION both aboard the ISS and on Earth, as the concentrated samples can be stored and transported in the detachable CFAC containers.

## 4. Conclusions

The iSSC prototype technology has been validated as microgravity-compatible, and the Phase III study has revealed its potential for in situ application on the ISS. It efficiently detects bacteria at concentrations of lower than 10^4^ CFU/L, which is significant, considering that the current ISS method of filtering 10 mL of potable water may not detect lower concentrations of biological contaminants. As such, for routine microbial monitoring using molecular assays, a concentration system like the iSSC, capable of filtering large-volume samples and efficiently delivering the concentrated microbes into a small sample volume, may be necessary. Furthermore, the iSSC system saves valuable crew time by concentrating a 1-L sample in about five minutes. The ability to filter large volumes of potable water (up to 5 L) aboard the ISS is crucial, especially in the event of contamination. Under such conditions, the iSSC system could be used to assess the microbial burden and also help with purifying the water by trapping the contaminants on the filter while concentrating large volumes of samples. Such “purification” would allow for the reuse of the water for washing and other maintenance tasks.

## Figures and Tables

**Figure 1 microorganisms-11-02310-f001:**
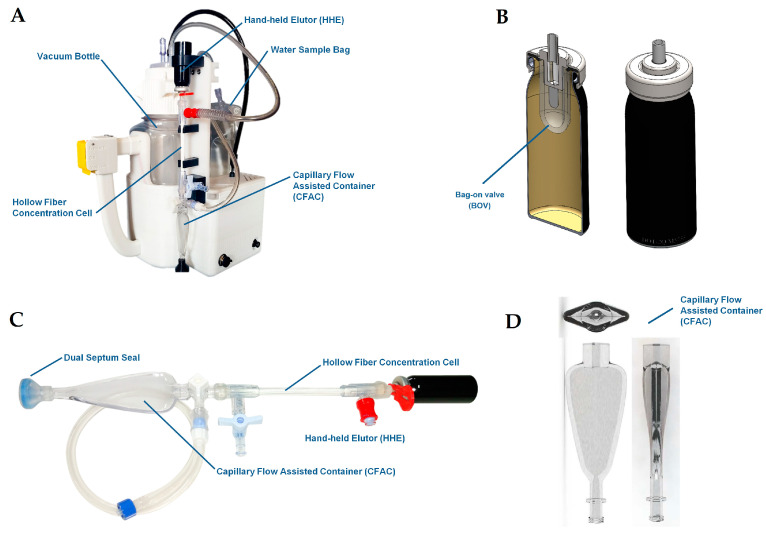
The iSSC components: iSSC overview (**A**); hand-held elutor (HHE) and bag-on-valve (BOV) assemblies (**B**); iSSC consumable assembly (**C**); and orthographic view of the capillary flow assisted container (CFAC) design (**D**). This figure was adapted from Urbaniak et al., 2020 [13].

**Figure 2 microorganisms-11-02310-f002:**
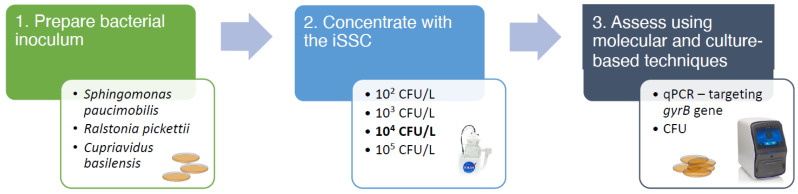
The overview of the experimental steps in phase III of the iSSC validation study. In the phase III study, three microorganisms *S. paucimobilis*, *C. basilensis* and *R. pickettii* in various concentrations 10^2^–10^5^ CFU/L were concentrated using the iSSC system. The concentration efficiency was assessed using culture-based and molecular-based approach using QuantStudio 6 system for qPCR analysis targeting species specific *gyrB* gene. The figure was adapted from Urbaniak et al., 2020 [13].

**Figure 3 microorganisms-11-02310-f003:**
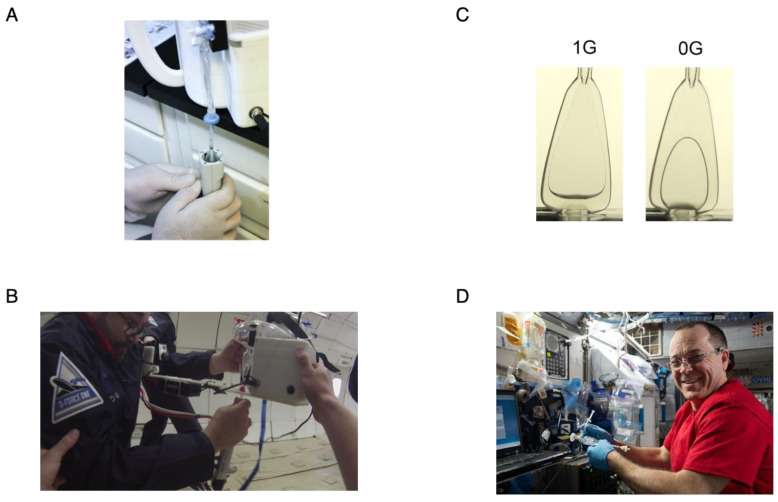
The validation of microgravity compatibility of the iSSC. CFAC sample extraction using repeater pipette (**A**); CFACs at 1 G and 0 G (**B**); iSSC operation during parabolic test flight (**C**); and the ISS flight test of the CFAC (**D**).

**Figure 4 microorganisms-11-02310-f004:**
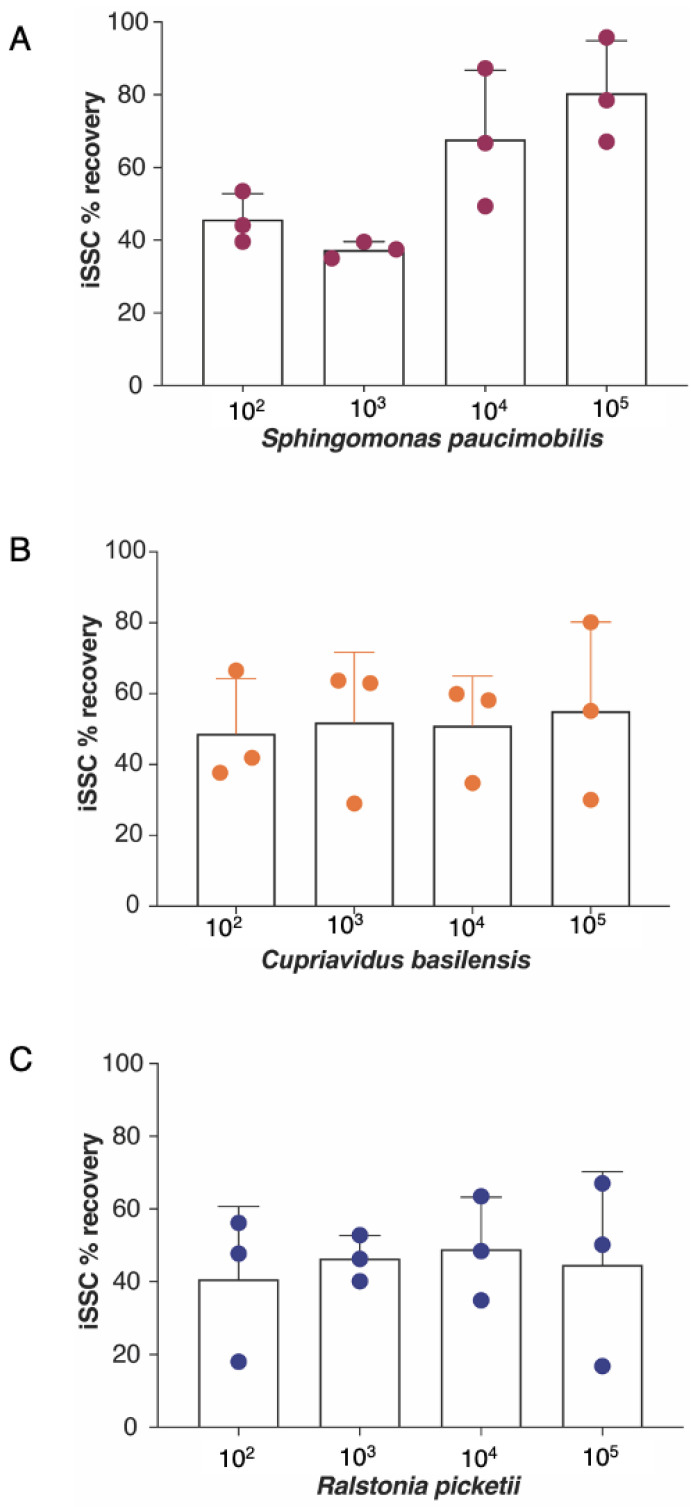
The iSSC concentration efficiencies, assessed with the culture-based method, for *S. paucimobilis* (**A**); *C. basilensis* (**B**); and *R. pickettii* (**C**). One liter of PBS containing 10^2^ to 10^4^ CFU/L of each bacterium was concentrated with the iSSC system. The concentrates and initial inoculum samples were plated on TSA plates and the CFUs were counted. The iSSC percent recovery efficiency was calculated by dividing the concentrated CFU counts by the original inoculum counts. Each point is a biological replicate that represents the mean of colony counts from two replicate plates. There were no statistically significant differences between the concentrations or strains.

**Figure 5 microorganisms-11-02310-f005:**
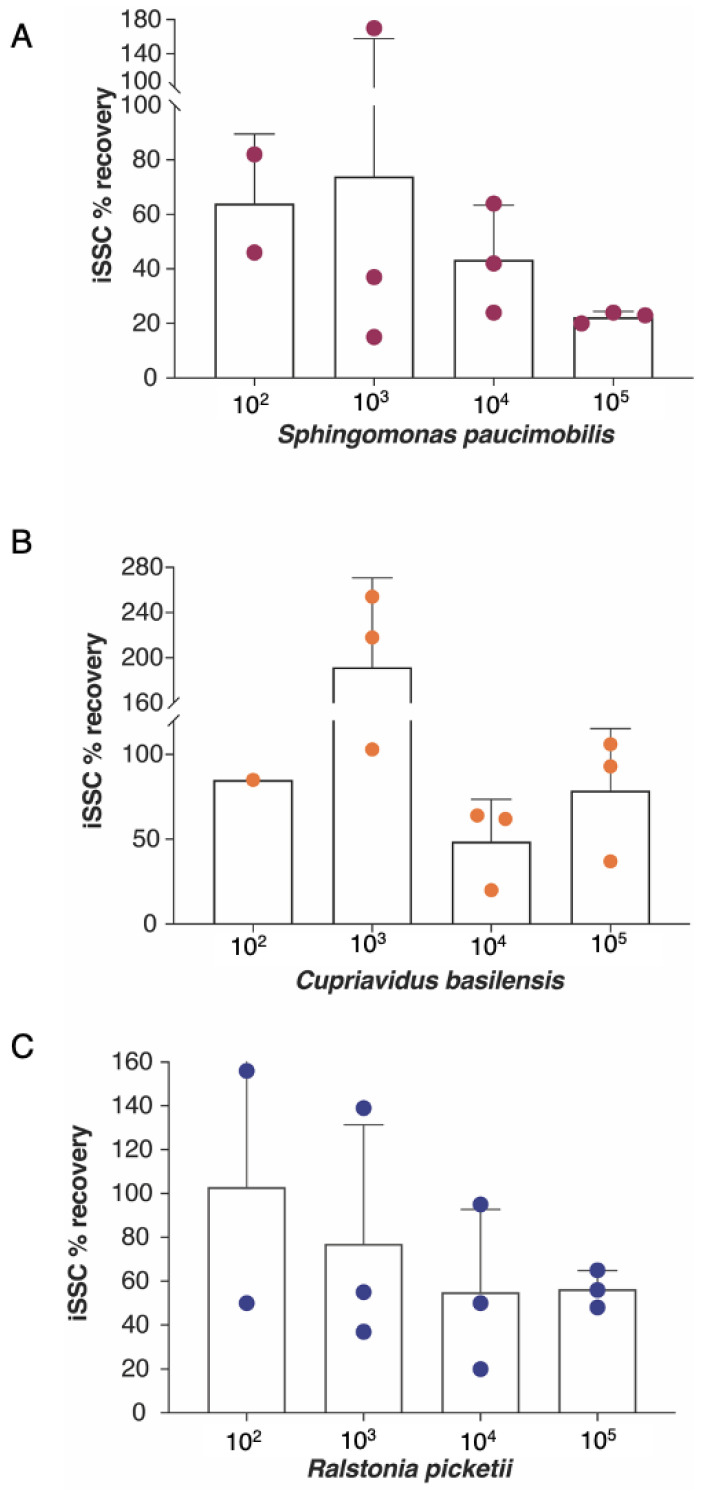
The iSSC concentration efficiencies assessed with the molecular-based method for *S. paucimobilis* (**A**); *C. basilensis* (**B**); and *R. pickettii* (**C**). After the concentration step and the assessment of the iSSC recovery with culturing, the samples were used for qPCR analysis. Each concentrated sample and corresponding inoculum were assessed for iSSC percent recovery efficiency using species-specific *gyrB* gene copy numbers. Biological and technical replicates were used to assess the reproducibility. Each point is a biological replicate that represents the average of three technical replicates. The copy number efficiency was reported as a percentage, and it was calculated by dividing the *gyrB* gene copy number of the concentrated sample by the *gyrB* gene copy number of the inoculum. Samples containing 10^2^ CFU/L were at the limit of detection of the qPCR assay performed using Quant6 Studio and, hence, were not always detected. There were no statistically significant differences between the concentrations or strains.

**Figure 6 microorganisms-11-02310-f006:**
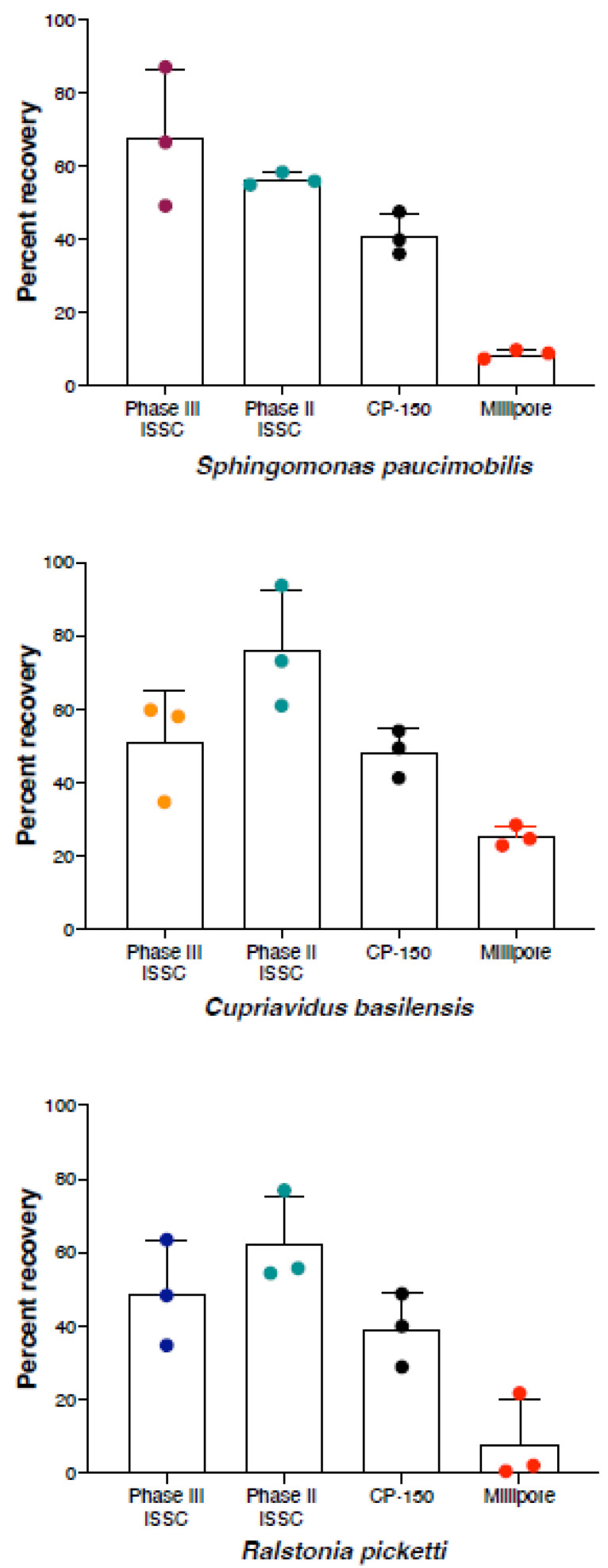
Comparison of the phase II and phase III results of the iSSC validation studies. The iSSC recovery is presented for *S. paucimobilis*, *C. basilensis*, and *R. pickettii*. During phase II, samples containing 10^4^ CFU/L were concentrated using iSSC, CP-150, and Millipore devices. Each point is a biological replicate that represents the mean colony count from at least two replicate plates.

**Table 1 microorganisms-11-02310-t001:** Summary of sequences for *C. basilensis*, *S. paucimobilis*, and *R. picketii gyrB* gene-specific primers, probes, and standards for qPCR analysis.

Primer	Organism	Sequence
SP_F	*Sphingomonas paucimobilis*	5′ CTG GAG TGG AAT GAC AGC TAT T 3′
SP_R		5′ GAC TTG TCG GCA TAA TTG TTG AG 3′
SP_probe		5′ CGT TCA CCA ACA ACA TCC CGC AG 3′
SP_std		5′ ATG GAT GTC GCG CTG GAG TGG AAT GAC AGC TAT TAT GAG AAC GTC CTG GCG TTC ACC AAC AAC ATC CCG CAG CGC GAC GGC GGC ACG CAT ATC GCG GCC TTC CGC GCG GCG TTG ACC CGC ACG CTC AAC AAT TAT GCC GAC AAG TCG GGC CTT CTG AA 3′
CM_F	*Cupriavidus basilensis*	5′ TGC TGC TGA CGT TCT TCT ATC 3′
CM_R		5′ CTC GTT GTC GTC CTT GAT GT 3′
CM_probe		5′ CGC TCT ACA AGA TCA AGC ACG GCA 3′
CM_std		5′ ATC CGC ACA CTG CTG CTG ACG TTC TTC TAT CGC CAG ATGC CGG ACA TCA TCG AGC GCG GCT ACG TGT ACA TCG CCC AGC CGC CGC TCT ACA AGA TCA AGC ACG GCA AGG AAG AGC GCT ACA TCA AGG ACG ACA ACG AGC TGA ACG CCT A 3′
RP_F	*Ralstonia picketii*	5′ TGC TGC TCA CGT TCT TCT AC 3′
RP_R		5′ GCC ATC TCG ACA TCG TCT TT 3′
RP_probe		5′ CGC TCT ACA AGA TCA AGC ACG GCA 3′
RP_std		5′ ACA TCC GCA CGC TGC TGC TCA CGT TCT TCT ACC GCC AGA TGC CCG AGA TCA TCG AGC GCG GCC ACG TGT ACA TCG CCC AGC CGC CGC TCT ACA AGA TCA AGC ACG GCA AGG AAG AGC GCT ACA TCA AAG ACG ATG TCG AGA TGG CCG CCT ACC TCG T 3′

“F” is the forward primer, “R” is the reverse primer and “std” refers to the standard. The specs for each probe were the same and consisted consisted of a fluorophore 6-FAM and quencher ZEN/Iowa Black FQ combination.

**Table 2 microorganisms-11-02310-t002:** The post-concentration recovery volumes for *S. paucimobilis*, *C. basilensis* and *R. pickettii* samples.

	Concentration CFU/L	*Sphingomonas paucimobilis* CFAC Volume [µL]	*Cupriavidus basilensis* CFAC Volume [µL]	*Ralstonia pickettii* CFAC Volume [µL]
Biologic replicate 1	10^2^	450	870	350
	10^3^	430	400	550
	10^4^	370	530	450
	10^5^	400	490	400
Biologic replicate 2	10^2^	680	890	400
	10^3^	370	350	400
	10^4^	390	330	220
	10^5^	430	510	350
Biologic replicate 3	10^2^	430	510	510
	10^3^	290	260	480
	10^4^	580	450	330
	10^5^	570	300	700
Average volume [µL]		449	491	428

**Table 3 microorganisms-11-02310-t003:** The comparison of mean volumes and concentration factors between the Phase II and Phase III validation studies.

	Millipore	Phase II iSSC	Phase III iSSC
Mean Volume (µL) for 10^4^ CFU/L	928	304	405
Concentration Factor	1077	3289	2469

## Data Availability

No data available to disclose. All are provided here or in the Appendix A.

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
