# Peer review of "Microbial Detection and Quantification of Low-Biomass Water Samples Using an International Space Station Smart Sample Concentrator"

_microorganisms, 2023, doi:10.3390/microorganisms11092310_

Round 1

Reviewer 1 Report

This research manuscript explored the role of ISS Smart Sample Concentrator as sample concentration method. It is acceptable in this this journal. After reviewing the content, I would recommend some suggestions to improve the quality of this manuscript.

Abstract, Introduction sections and figures (4 and 5): Instead, 10^2 write 102 (2 as superscript)

Abstract section: Replace ~450 µL with 450 µL

Line 46: Correct the citation (NASA, ed, 2003)

Abstract and Introduction sections: Replace, >1L with >1 L (make space)

Discussion section: It will be good if authors discuss the inability of qPCR assay to differentiate between alive and dead cells.

There are several typographic and grammatical errors.

Line 91: Correct accordingly n.d.). ventured

Line 184: Correct thee with the

Author Response

Abstract, Introduction sections and figures (4 and 5): Instead, 10^2 write 102 (2 as superscript) Corrected

Abstract section: Replace ~450 µL with ≈ 450 µL Corrected

Line 46: Correct the citation (NASA, ed, 2003) Corrected

Abstract and Introduction sections: Replace, >1L with >1 L (make space) Corrected

It will be good if authors discuss the inability of qPCR assay to differentiate between alive and dead cells.
Thank you for the suggestion please see the added sentence:
"
It is also worth mentioning that in this study dead and alive cells were not distinguishable via qPCR as no intercalating dye like propidium monoazide (PMA) was added to the concentrated samples."

Line 91: Correct accordingly n.d.). ventured Corrected

Line 184: Correct thee with the Corrected

Reviewer 2 Report

The paper presented by the authors shows a system for quantifying and detecting microorganisms in low biomass water samples using the intelligent sample concentrator of the International Space Station. They give a critical issue to ensure the quality of water on board for astronauts; they offer a system that seeks to detect non-culturable microorganisms in water using a system that can quickly and efficiently concentrate large-volume samples. 

The system presented by the authors is interesting in the detection and quantification processes of water bodies' microbial quality, which could be projected to other scenarios where it is crucial to determine water quality. It is also a system that could guarantee the reuse of water resources. 

The following are some observations.

1. In the introduction, the authors could give a context of the different methods of detecting water microorganisms and the advantages and disadvantages of this one.

2. It is understood that the system presented by the authors is focused on the existing conditions on board a ferry; the contribution of this work to the knowledge and technologies for the detection of microorganisms in bodies of water must be expanded.

3. In the methodology, the order of the figures should be revised; Figure 3 appears after Figure 4, which makes it difficult to understand what is established in the method.

4. It is suggested that the authors present the physicochemical characteristics of the water used since these characteristics affect the growth or not of the microorganisms.

5. In line 163, the authors indicate, "There were no statistically significant differences between the concentrations or strains" Reviewing the figures, this statement is not evident; for example, in the Figure of S. paucimobilis, it can be seen that there are differences between concentrations about the iSSC % recovery. The GraphPad Prism program allows for establishing if there are significant differences; it is suggested in the program to perform the ANOVA and place the existence of significant differences between each one of the concentrations in the bars. Another option is that the authors, if possible, set the tables resulting from the statistical analysis to determine the existence of significant differences that support the assertion.

6. Make the adjustments in figures 5 and 6 in the previous comment.

7. It is suggested to make graphs showing each phase's efficiency by the three microorganisms.

Author Response

1. In the introduction, the authors could give a context of the different methods of detecting water microorganisms and the advantages and disadvantages of this one.
On the ISS microbes in water are detected via cultivation, the use of iSSC skips that step allowing to use of molecular methods instead. The authors discussed concentration methods that were more relevant to the presented device. Lines 55-62

2. It is understood that the system presented by the authors is focused on the existing conditions on board a ferry; the contribution of this work to the knowledge and technologies for the detection of microorganisms in bodies of water must be expanded.
The iSSC system is designed to tackle issues related to microgravity. The CP150 and CP Select are good for concentrating other aquatic samples.  This was stated in Line 64 to 66. 

3. In the methodology, the order of the figures should be revised; Figure 3 appears after Figure 4, which makes it difficult to understand what is established in the method.
Thank you, the order was adjusted.

4. It is suggested that the authors present the physicochemical characteristics of the water used since these characteristics affect the growth or not of the microorganisms.

As stated in Line 154 to 157, we used phosphate buffered saline (pH 7.4) and hence there are no physicochemical properties that we could present. 

5. In line 163, the authors indicate, "There were no statistically significant differences between the concentrations or strains" Reviewing the figures, this statement is not evident; for example, in the Figure of S. paucimobilis, it can be seen that there are differences between concentrations about the iSSC % recovery. The GraphPad Prism program allows for establishing if there are significant differences; it is suggested in the program to perform the ANOVA and place the existence of significant differences between each one of the concentrations in the bars. Another option is that the authors, if possible, set the tables resulting from the statistical analysis to determine the existence of significant differences that support the assertion.

Appreciate the suggestions. Unfortunately, getting the GraphPad license renewed is taking time. Since there is no significance, we added this information in the Figure 5 (See Lines 366 to 377). If the Editor need this to be fixed, then there will be a delay in submitting the review until mid Sep.

6. Make the adjustments in figures 5 and 6 in the previous comment.

Same explanations as given in query #5.

7. It is suggested to make graphs showing each phase's efficiency by the three microorganisms.

The first bar is for Phase III and subsequent bars (2nd to 4th) are for Phase II. The following sentences in Line 369 to 372 mention this.

During phase II samples containing 104CFU/L were concentrated using iSSC, CP-150, and Millipore devices. Each point is a biological replicate that represents the mean of colony counts from at least two replicate plates.